# Microbiota Gut–Brain Axis in Ischemic Stroke: A Narrative Review with a Focus about the Relationship with Inflammatory Bowel Disease

**DOI:** 10.3390/life11070715

**Published:** 2021-07-19

**Authors:** Emanuele Sinagra, Gaia Pellegatta, Valentina Guarnotta, Marcello Maida, Francesca Rossi, Giuseppe Conoscenti, Socrate Pallio, Rita Alloro, Dario Raimondo, Fabio Pace, Andrea Anderloni

**Affiliations:** 1Endoscopy Unit, Fondazione Istituto San Raffaele—G. Giglio, Contrada Pietra Pollastra Pisciotto, 90015 Cefalù, Italy; fraross76@hotmail.com (F.R.); dottgconoscenti@gmail.com (G.C.); ritalloro@hotmail.it (R.A.); dario.raimondo@hsrgiglio.it (D.R.); 2Euro-Mediterranean Institute of Science and Technology (IEMEST), 90100 Palermo, Italy; 3Digestive Endoscopy Unit, Division of Gastroenterology, Humanitas Research Hospital, 20089 Rozzano, Italy; gaia.pellegatta@humanitas.it (G.P.); andrea_anderloni@hotmail.com (A.A.); 4Endocrinology Section, PROMISE Department, AOUP Paolo Giaccone, 90127 Palermo, Italy; valentinaguarnotta@gmail.com; 5Gastroenterology and Endoscopy Unit, S. Elia-Raimondi Hospital, 93100 Caltanissetta, Italy; marcello.maida@hotmail.it; 6Endoscopy Unit, Department of clinical and experimental medicine, University of Messina, AOUP Policlinico G. Martino, 98125 Messina, Italy; socratep@tin.it; 7Emergency Unit, Fondazione Istituto G. Giglio, Contrada Pietra Pollastra Pisciotto, 90015 Cefalù, Italy; 8Unit of Gastroenterology, Bolognini Hospital, 24100 Bergamo, Italy; fabio.pace@unimi.it

**Keywords:** gut, brain, microbiota, stroke, inflammatory bowel disease

## Abstract

The gut microbiota is emerging as an important player in neurodevelopment and aging as well as in brain diseases including stroke, Alzheimer’s disease, and Parkinson’s disease. The complex interplay between gut microbiota and the brain, and vice versa, has recently become not only the focus of neuroscience, but also the starting point for research regarding many diseases such as inflammatory bowel diseases (IBD). The bi-directional interaction between gut microbiota and the brain is not completely understood. The aim of this review is to sum up the evidencesconcerningthe role of the gut–brain microbiota axis in ischemic stroke and to highlight the more recent evidences about the potential role of the gut–brain microbiota axis in the interaction between inflammatory bowel disease and ischemic stroke.

## 1. Introduction: The Gut–Brain Microbiota Axis

The extent and mechanisms of interaction between the gut microbiota, defined as an ecological unit composed of microorganisms within a specific environment, and non-gastrointestinal organs are still scanty, although many experimental and clinical experiences show that a bidirectional communication exists between the gut and its microbiota, and the brain [1]. This system, which has not been entirely explored, is based on neural, endocrine, immunological and metabolic pathways [2,3,4]. The communication between the brain and the gut occurs through both neuronal and non-neuronal mechanisms [1]. These communications are defined as “top-down” when signaling is directed from brain to gut, and defined as “bottom-up” when signaling is directed from gut to brain.

In the “bottom up” signaling, the central nervous system (CNS) receives input through a number of different mechanisms. The CNS includes the hypothalamus, amygdala, and hippocampus and their interaction with emotional centers localized within the limbic system, which are mainly involved in the control of body reaction in response to stress [2]. These communications occur:Directly, through the vagus nerve (VN), whose stimulation is mediated by microbial metabolites and neuro-hormones [1] released from the enteric nervous system (ENS), that controls bowel function even though it is completely separate from the CNS; the ENS is made up of interneurons, sensory neurons, motor neurons, and neurotransmitters [2]. Furthermore, neuroinflammation could be caused also by the production of immunogenic microbial endotoxins (such as LPS), that act both through a direct damage or through the activation of immune cells [1,5,6].Indirectly, through the microbial releasing of metabolites such as short-chain fatty acids (SCFA), bile acids, indoles and neurotransmitters that, after entering the systemic blood, travel to the brain in order to modulate the function of neurons, microglia, astrocytes, and the blood brain barrier [1,7,8,9].In the “top-down” signaling, the gut microbiota receives input:-Indirectly, from the enteric nervous system (ENS). In this context, also the neuroendocrine signaling network mediated by the hypothalamic–pituitary–adrenal (HPA) axis, activated by the integrative reactions of specific centres in the CNS, plays a pivotal role; in fact, it represents a central integrative system essential for the successful physiological adaptation of our organism to stress [2].-Directly, through the autonomic nervous system (ANS), the pivotal modulator of the ENS [2].



Based on this premise, it is conceivable that gut microbiota might play an important role both under pathological conditions and in physiological processes [10]. Researchers have identified changes in gut microbiota composition of in several diseases, such as inflammatory bowel disease [11], irritable bowel syndrome [12], diabetes [13], cancer [14], and diseases of the nervous system, such as Alzheimer’s disease [15], Parkinson’s disease (PD) [16], spinal cord injury [17], autism [18], and stroke [19].

The aim of this narrative review is to sum up the evidence about the role of the gut–brain microbiota axis in ischemic stroke and to highlight the more recent evidence about the potential role of the gut–brain microbiota axis in the interaction between IBD and ischemic stroke.

## 2. The Role of Microbiota Gut–Brain Axis in Ischemic Stroke

According to the World Health Organization, strokes are the second leading cause of death and the third leading cause of disability worldwide [20]. Ischemic stroke (IS), defined as an obstruction within a blood vessel supplying blood to the brain (most frequently to the middle cerebral artery (MCA)), accounts for about 70–80% of all strokes [21].

Alterations in gut microbiome can be a risk factor and may also lead to IS. Risk factors for IS and alteration of gut-microbiome composition (defined as dysbiosis) are influenced by similar factors, including aging, metabolic diseases, hypertension, and vascular dysfunction [22] Furthermore, the systemic inflammatory response after IS can impair the clinical outcome after IS, therefore yielding to liver, renal, respiratory, gastrointestinal, and cardiovascular impairment, including the multiple organ dysfunction syndrome [22]. Figure 1 summarizes the signaling involved in the microbiota brain–gut axis.

### 2.1. The Role of Gut Dysbiosis inInfluencing Stroke Risk Factors

#### 2.1.1. Aging

Little is known about how the gut–brain axis changes with aging [23]. Aging is associated with an impairment of the gut epithelial barrier, a loss of enteric neurons, and an altered mucosal immune function that cause an imbalance in the secretion of proinflammatory cytokines [1]. Such changes, often defined as “inflammeging”, may decreases the ability in the elderly to cope with antigenic, toxic, physical and ischemic stress [1].

Recently, Lee et al. [19] performed a systematic review including eighteen studies, and suggested that aging, inflammation, and different microbial compositions could contribute to ischemic stroke. Interestingly, it was found that the aged mice with a high Firmicutes/Bacteroidetes ratio were more exposed to an IS, and unable to recover from neurological deficits, thus showing a higher mortality rate [23,24].

#### 2.1.2. Metabolic Diseases

With regard to the role of metabolic diseases, evidence arising from randomized controlled trials, showed that lean-donor fecal microbiota transplantation (FMT) in subjects with metabolic syndrome improved obesity and insulin sensitivity, thus suggesting that the microbiota can regulate host metabolism as an “organ” [25,26].

It is already known the effect of microbiota-derived metabolites of carbohydrates and proteins on host metabolism. The importance of the involvement of gut–brain microbiota axis in controlling host metabolism was also demonstrated from experimental studies showing that acetate, a SCFA, reaches the hypothalamus through the blood-brain barrier, and induces the production of gamma-aminobutyric acid, resulting in suppression of central appetite [25,27].

Further pathways include also: peptide YY production through GPR41, which inhibits intestinal motility and increases the absorption rate of nutrients through the intestinal epithelium [28], the beneficial role of indole and its derivatives (through the production of the incretin hormone GLP-1 from intestinal enteroendocrine cells) [29] and the harmful effect of imidazole propionate, a microbial metabolites of histidine [25,30]. Furthermore, microbial-derived metabolites control bile acid homeostasis via farnesoid X receptors, which can also influence glucose metabolism [25,31].

#### 2.1.3. Arterial Hypertension and Vascular Dysfunction

The gut microbiota is probably involved in the genesis of hypertension, despite the mechanism is not yet fully elucidated.

It has been shown that in spontaneously hypertensive rats, a significant decrease in the composition of microflora in the gut occur, associated with an increase in the ratio of Firmicutes/Bacteroidetes [32,33]. Infusion of angiotensin II (AngII) attenuated the blood pressure increase in germ-free mice compared with conventionally raised mice, indicating that gut microbiota involves blood pressure regulation [32,34].

Furthermore, the role of various G-protein-coupled receptors (GPRs) in hypertension has been pointed out [32,35]. In fact, the gut microbial metabolites SCFAs modulate the activity of GPRs, including GPR41, GPR43, and GPR109A [32,35].

On the other hand, high-fiber diet and acetate supplementation significantly decrease diastolic blood pressure, cardiac fibrosis and ventricular hypertrophy when compared to a control standard diet [36] thus suggesting that the gut microbiota-producing SCFAs in circulation play an important role in hypertension [32].

Finally, the gut microbial metabolite, SCFAs, and their effect on ox-LDL levels and other pathways may also contribute to increasing arterial pressure [32], for example by inhibiting nitric oxide synthesis (NO) and endothelin-1 [37,38].

As in the case of arterial hypertension and metabolic diseases, bacterial metabolites such as SCFAs, nitrites, flavanol, Trimethylamine N-oxide (TMAO), indoles, and sulfidic acid have been identified all as causative factors of vascular dysfunction [22]. In the case of TMAO, a link between this microbial metabolite with atherosclerosis has been well established [1]. SCFAs and sulfidic acid are particularly capable of inducing vasodilatation, whereas indole and TMAO increase the production of reactive oxygen species, thus reducing cerebral vasodilatation [22].

### 2.2. The Effect of Gut Dysbiosis in Influencing Stroke Outcomes

As indicated by the aforementioned review performed by Lee et al. [19], stroke may be linked with gut dysbiosis, altering the microbial composition, and therefore conditioning the post-stroke outcome [6,23,39,40,41,42,43,44,45,46,47,48,49,50,51,52,53]. An inverted Firmicutes/Bacteroidetes ratio is suggested to be a hallmark for aging, and it is significantly associated with ischemic stroke in mouse models [19].

In fact, these studies focused on dysbiosis-induced intestinal paralysis, an increased gut permeability and the loss of cholinergic innervation in ileum, and an increased sympathetic activity [1,5,40,54].

Another point of interest is the relationship between the dysbiosis and specific inflammatory markers that are present in the stroke mice. As SCFAs and TMAO have been shown as the emerging factors in ischemic stroke in animal studies, further studies investigating the association between SCFAs and TMAO among patients need to be performed [19].

Furthermore, changes caused by stroke on the gut microbiota can induce neurological complications, stroke-associated pneumonia, cardiovascular complications, gastrointestinal complications, and renal dysfunction, with possible development of the systemic inflammatory response and multiple organ dysfunction syndromes [22].

With regard to neurological complications, after stroke, innate immune cells respond within hours, followed by the adaptive immune response through activation of T and B lymphocytes. Subpopulations of T-cells can help or worsen ischemic brain injury [21]. Pro-inflammatory Th1, Th17, and γδ T-cells are often associated with increased inflammatory damage, whereas regulatory T-cells are known to suppress post-ischemic inflammation by increasing the secretion of anti-inflammatory cytokine IL-10 [21]. Therefore, all these changes could influence the stroke severity, and therefore the cognitive, motor and sensory dysfunction.

Complication following stroke is pneumonia, representing a further example of dysbiosis-induced infection, with an incidence occurring in up to 10% of cases of IS [22,55], and in turn linked with immunosuppression and dysphagia [22,55,56].

Cardiovascular complications, occurring in up to 39% of cases of IS, and including several conditions, such as arrhythmias, Takotsubo syndrome, and myocardial infarction, are presumably linked with post-stroke dysbiosis [22,57]. In fact, as aforementioned, in this setting the altered secretion of TMAO increases the production of reactive oxygen species, thus reducing cerebral vasodilatation. Moreover, other studies demonstrated that serum levels of gamma-butyrobetaine and trimethyl-lysine, metabolites of carnitine, were also associated with cardiovascular death [58]. With regard to renal dysfunction, occurring in up to 10% of cases of IS, it is also influenced by stroke-induced dysbiosis [10,59]. Here, the link is atherosclerosis [60]; and probably also the altered secretion of TMAO following an inflammatory process which supports the existence of a gut–brain–kidney axis, associated with tubule-interstitial fibrosis and renal dysfunction [10,58,61].

Finally, a multiple organ dysfunction syndrome (MODS), due to the release of intestinal lumen bacteria and toxins into the circulation, may occur in up to 12% of cases of IS thus enhancing systemic inflammation and causing sepsis or a systemic inflammatory response [62]. Despite that a clear correlation between its occurrence and a microbiota gut–brain axis has not been clearly demonstrated, an impaired intestinal permeability (defined as “leaky gut”) could be considered as trigger factor in this setting [10].

## 3. The Role of the Gut–Brain Microbiota Axis in the Occurrence of Gastrointestinal Complications after Ischemic Stroke, and the Focus about the Link between Inflammatory Bowel Disease and Ischemic Stroke

After an acute ischemic stroke up to 50% of patients can present gastrointestinal complications such bleeding, bowel obstruction and incontinence, and dysphagia. These complications may increase the days of hospitalization and the risk of mortality [51]. To date, clinical and experimental data about gastrointestinal consequences after acute ischemic stroke are limited (Table 1).

Experimental traumatic brain injury in rats is associated with severe mucosal atrophy and disruption of gut epithelial cell tight junctions after few hours from stroke, which can persist for 7 days [60].

In two Chinese studies it has been observed in murine models that ischemic stroke leads to destruction and necrosis of the small bowel villus epithelium with consequences on small intestinal motility [63,64].

Recently, it has been demonstrated that experimental induced stroke in mice leads to noradrenaline release with altered cecal mucoprotein production and goblet cell numbers. These changes are associated with a mutation in the composition of cecal microbiota, with specific changes in *Peptococcaceae* and *Prevotellaceae* correlating with the extent of injury [65].

In a recent prospective case–control study rectal swabs from stroke patients were collected within 24 h of hospital admission. This study showed an altered microbiota composition in adults with acute ischemic stroke and cerebral hemorrhage. In particular, patients after stroke showed a higher level of bacteria implicated in trimethylamine-N-oxide (TMAO) production and a loss of butyrate-producing bacteria. In particular, butyrate-producing bacteria have been identified to play a protective role against a variety of systemic infectious diseases. Moreover, a butyrate-producing bacteria increase was independently associated with reduced infection rate [44].

In a recent study the authors found a reduction of SCFAs-producing bacteria in cases with ischemic stroke when compared to controls [45]. Nonetheless, a study showed that the butyrate-producing bacteria were remarkably less abundant in ischemic stroke and with increasing abundance of lactic acid bacteria [46].

In another study the authors investigated the effects on cellularity of the lymphoid tissue of the gut. In particular, they observed a reduction of T and B cells after ischemic stroke, and a stability in the amount of natural killer cells and macrophages [66].

A reduction in parasympathetic nerve activity following acute CNS injury is associated with overgrowth of bacteria in the bowel and increased bacterial translocation. Substances released by bacteria and metabolites and media released by the gastrointestinal immune system can affect gastrointestinal movement [67,68].

In particular, it has been described that, after CNS injury as ischemic stroke, mast cells present in the gut release histamine and tryptase activating submucosal neurons and peptidergic neurons to secrete vasoactive intestinal peptide (VIP) which alters the movements of the gastrointestinal tract [69].

Enteric nervous system with the release of nitric oxide can also lead to a delay of gastric emptying [70].

Recently a link between ischemic stroke and inflammatory bowel disease (IBD) has been proposed and some interesting studies about the relationship between these two clinical entities are currently available in literature.

In particular, IBD patients present a higher risk for arterial and venous thromboem-bolism (TE) than patients without IBD and this complication is associated with a consid-erable morbidity and mortality rates (overall mortality 25% per episode). In particular, cerebrovascular TE represents the most frequent and severe CNS complication in IBD [70].

In a retrospective single center cohort study, it has been demonstrated that the risk of TE is higher in patients with active IBD than IBD in remission [71].

In a population-based retrospective cohort study it has been shown that the frequency of IBD exacerbation and hospitalization are risk factors for ischemic stroke [72].

Current data regarding cerebrovascular risk modification induced by treatment for IBD are contradictory. A beneficial effect with increased carotid-femoral pulse wave velocity (PWV) has been shown with salicylates, but not with steroids or azathioprine. Moreover, TNF-a inhibitors appear to decrease the ischemic heart disease rate but increase the risk of cerebrovascular events [73].

Currently some studies reported a possible role of gut dysbiosis in the association between IBD and ischemic stroke (Table 1).

In particular in IBD patients has been described an expansion of potential pathogens such as Enterobacteriaceae including *E. coli* and changes in microbial composition with a reduction of Firmicutes species [74]. Moreover, the metabolic pathway of amino acid biosynthesis, carbohydrate metabolism, oxidative stress and bile salt seem to be modified in the microbiota of patients affected by IBD, suggesting a functional impact of gut microbiota on pathogenesis of these diseases [74].

In IBD patients has been reported an aberrant immune response to microbial dysbiosis due to genetic alteration in innate immunity, intestinal barrier, microbial recognition and processing. It leads to a persistently stimulation of proinflammatory condition and macrophage/monocyte infiltration in the gut [3].

Some recent studies present an excellent example for the modulation by gut microbiota through gut–brain axis via bottom-up in IBD related neurological complication such as memory impairment and anxiety-like behavior in animal models [75,76,77].

Since IBD patients present higher risk of ischemic stroke, the gut–brain axis is very likely to represent a potential link between gut pathology and the increased risk of ischemic stroke. However, currently there are still no studies in the literature regarding the role of gut–brain axis in the relationship between ischemic stroke and IBD. Moreover, there is no data about the potential role of IBD therapies on risk of ischemic stroke.

Future studies are needed to establish causal association between active IBD and cerebrovascular events in order to also establish a possible therapeutic target.

## 4. Conclusions and Perspectives

The brain–gut axis communication occur through both neuronal and non-neuronal pathways supporting notion that gut microbiota might play a role in many pathological and physiological conditions. A relationship between gut microbiota and ischemic stroke has been reported. The gut dysbiosis has a strong influence on stroke risk factors such as aging, metabolic disease and arterial hypertension. In addition, the brain–gut axis appear significantly distressed after stroke by injury, with induced damage-associated molecular patterns (DAMPs) and cytokine release, resulting in neurological, cardiovascular, gastrointestinal and nephrological complications with a possible multiple organ dysfunction. Notably, the gastrointestinal complications are very common and may be related to an increase in bacteria involved in the TMAO production and loss of butyrate-producing bacteria. Interestingly, a link between inflammatory bowel disease and its grade activity with ischemic stroke has been suggested. However, the exact molecular mechanisms underlying the changes in the brain–gut axis and the related inflammatory and immune responses after ischemic stroke should be further investigated. Prospective studies to identify the bacterial species of microbiota involved in ischemic stroke are required. Gut microbiota analysis could act as a potential personalized therapeutic approach for the treatment of cardiovascular and metabolic diseases against stroke.

## Figures and Tables

**Figure 1 life-11-00715-f001:**
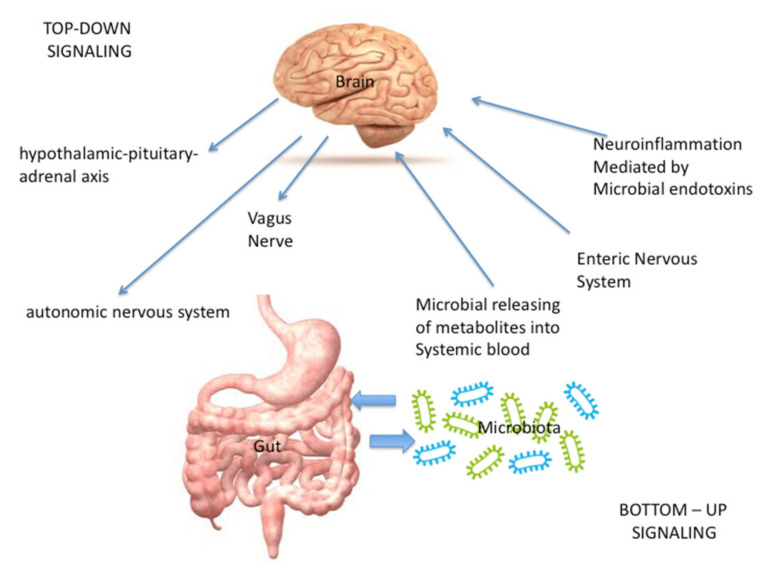
Summarizes the signaling involved in the microbiota brain–gut axis.

**Table 1 life-11-00715-t001:** Summarizes the evidences regarding the role of the gut–brain microbiota axis in the occurrence of gastrointestinal complications after ischemic stroke, and the link between the microbiota gut–brain axis, IBD and ischemic stroke.

Role of the Gut–Brain Microbiota Axis in the Occurrence of Gastrointestinal Complications after Ischemic Stroke				
Author	Year of Publication	Type of Study	Population	Results
Hang C.H. et al. [64]	2003	Pre-clinical study	Mice subjected to experimental stroke	Severe mucosal atrophy and disruption of gut epithelial cell tight junctions after few hours from stroke to 7 days
Xu X. et al. [63]	2012	Pre-clinical study	Mice subjected to experimental stroke	Decreased gastrointestinal motility and damage to the intestinal mucosa existed in rats with experimental stroke
Liu Y. et al. [65]	2017	Pre-clinical study	Mice subjected to experimental stroke	Ischemic stroke significantly damaged the intestinal epithelium and activated intestinal immunity.
Houlden A. et al. [66]	2016	Pre-clinical study	Mice subjected to experimental stroke	Noradrenaline release with alteration of cecal mucoprotein production, goblet cell numbers, composition of cecal microbiota, with specific changes in *Peptococcaceae* and *Prevotellaceae*
Haak B.W. et al. [42]	2020	Prospective case–control study	Stroke patients and controls	Altered microbiota composition in adults with stroke with higher level of bacteria implicated in trimethylamine-N-oxide (TMAO) production and a loss of butyrate-producing bacteria.
Tan C. et al. [43]	2021	Prospective case–control study	Stroke patients and controls	Dysbiosis of SCFAs-producing bacteria and SCFAs in AIS patients increased the subsequent risk for poor functional outcomes
Li H. et al. [44]	2020	Prospective case–control study	Stroke patients and controls	The abundance and functions ofbutyrate-producing bacteria in stroke patients were significantly decreased while lactic acid bacteria were increased.
Schulte-Herbrüggen O. et al. [67]	2009	Pre-clinical study	Mice subjected to experimental stroke	Peyer’s patches revealed a significant reduction of T and B cell counts after cerebral ischemia.
**Link between inflammatory bowel disease and ischemic stroke**				
Bollen L. et al. [73]	2016	retrospective monocentric cohort study	IBD patients with a history of TE	83% of patients developed a venous TE. At the time of TE, 71% patients were diagnosed with active disease
Huang W.S. et al. [45]	2014	population-based retrospective cohort study	IBD adult patients and IBD-free controls	The risk of ischemic stroke was 1.12-fold higher among the IBD cohort than among the non-IBD cohort. The risk of developing ischemic stroke significantly increased with the increased frequency of IBD exacerbation and hospitalization.

## Data Availability

Not applicable.

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
