# Peer review of "Microbiota Gut–Brain Axis in Ischemic Stroke: A Narrative Review with a Focus about the Relationship with Inflammatory Bowel Disease"

_life, 2021, doi:10.3390/life11070715_

Round 1
Reviewer 1 Report
The concept and title of this review “Microbiota gut-brain axis in ischemic stroke: a narrative review with a focus about relationship with inflammatory bowel disease” is interesting and novel. Unfortunately, the subject of the manuscript given in the title is not adequately addressed. In general, the Authors present the association between gut dysbiosis and ischemic stroke and between ischemic stroke and inflammatory bowel disease (IBD). However, the role of gut microbiota in the context of association between IBD and ischemic stroke is not presented, whereas that should be the main focus of this paper. Regarding some recent comprehensive reviews on the role of gut dysbiosis in ischemic stroke, the current paper is missing novelty. Moreover, many aspects of the field are presented in rather superficial and even confusing way. Taking also into account more specific comments listed below, regretfully, I do not find this paper suitable for publication in its current form.
Specific comments:
- Some chaotic way of writing is reflected even by the use of term “microbiota-gut-brain axis”. The authors should decide if the will use bottom-up (microbiota-gut-brain axis) or top-down (brain-gut-microbiota axis) approach and then use it consequently, with proper use of dashes. While in the current text different modifications of this term occur (with or without dashes that is changing the meaning). The term “gut-brain microbiota axis” does not make sense. Sometimes even in one sentence, two different terms coexist evoking confusion – see page 6 – Table 1 title: “the gut-brain microbiota axis” and “ microbiota gut-brain axis”.
- In the Introduction the bottom-up and top-down signaling should be clearly distinguished, while, probably by mistake, the top-down signaling is mentioned just as the next bullet point.
- In the Introduction there some unnecessary repetitions and incorrect simplifications (e.g. the ENS consists also from glial cells that outnumber neurons).
- In general the titles of figures should be modified (It should not be: “Figure one summarizes….).
- Figure 1 needs to be corrected. The vagus nerve should not be presented as a key part of top-down signaling, since it comprises mainly afferent nerves (up to 90%). Please see Figure 1 in Lee et al., 2021 (ref. 22).
- It is difficult to agree with the first statement of paragraph 2a (page 3) that “little is known about how the gut-brain axis changes with aging”. Actually, there are numerus data on age-related changes in gut microbiota characterized by its decreased stability and diversity.
- Page 3, paragraph 2a – Metabolic syndrome as a risk factor for ischemic stroke is associated with well characterized changes in gut microbiota composition that should be clearly presented.
- In paragraph 2b – “The effect of gut dysbiosis in influencing stroke outcomes” – the causal relationship between ischemic stroke and gut dysbiosis is not clearly presented. Potential mechanisms of stroke-induced dysbiosis should be highlighted, whereas the Authors are pointing to dysbiosis-induced intestinal paralysis and increased gut permeability. This is confusing in understanding of the cause and consequence effect.
- Further, on page 6, the well know association between IBD and ischemic stroke is briefly highlighted, but nothing is mentioned about the role of gut dysbiosis in that association. Again, according to the main title, that should be the main focus of this narrative review.
- Table 1 distinctly demonstrates lack of flow of the review. It consists of two separate parts: 1) the role of microbiota-gut-brain axis in stroke related gastrointestinal symptoms and 2) link between IBD and ischemic stroke. By the way, it is not clear why only 4 papers were selected for the first part of the table, since there are much more data on that topic – see ref. 24.
Minor comments:
- There are multiple reference repetitions – ref. 1=67, ref. 5=42=66, ref. 8=9, ref. 19=24.
- Editing of English language is required. Just to give one example: in the Abstract, 3rd line from the bottom – it should be “evidence” instead of “evidences”.
Author Response
Response to reviewers’ comments
Dear Editors, dear Reviewers,
We wish to express our appreciation to the Editors and Reviewers for their insightful comments, which have helped us significantly to improve our manuscript. According to the suggestions, we have thoroughly revised our manuscript and its final version is enclosed. Point-by-point responses to the comments are listed below.
Reviewers’ comments #1
General comments: “The concept and title of this review “Microbiota gut-brain axis in ischemic stroke: a narrative review with a focus about relationship with inflammatory bowel disease” is interesting and novel. Unfortunately, the subject of the manuscript given in the title is not adequately addressed. In general, the Authors present the association between gut dysbiosis and ischemic stroke and between ischemic stroke and inflammatory bowel disease (IBD). However, the role of gut microbiota in the context of association between IBD and ischemic stroke is not presented, whereas that should be the main focus of this paper. Regarding some recent comprehensive reviews on the role of gut dysbiosis in ischemic stroke, the current paper is missing novelty. Moreover, many aspects of the field are presented in rather superficial and even confusing way. Taking also into account more specific comments listed below, regretfully, I do not find this paper suitable for publication in its current form.”
Response: We treated more extensively the relationship between IBD and ischemic stroke and we modified the manuscript accordingly, as in the revised and uploaded version of the manuscript
- Specific comments:
“Some chaotic way of writing is reflected even by the use of term “microbiota-gut-brain axis”. The authors should decide if the will use bottom-up (microbiota-gut-brain axis) or top-down (brain-gut-microbiota axis) approach and then use it consequently, with proper use of dashes. While in the current text different modifications of this term occur (with or without dashes that is changing the meaning). The term “gut-brain microbiota axis” does not make sense. Sometimes even in one sentence, two different terms coexist evoking confusion – see page 6 – Table 1 title: “the gut-brain microbiota axis” and “ microbiota gut-brain axis”.
Response: We corrected the whole manuscript by using the terms microbiota-gut-brain axis with dashes
“In the Introduction the bottom-up and top-down signaling should be clearly distinguished, while, probably by mistake, the top-down signaling is mentioned just as the next bullet point.”
Response: Since it was a typo, we corrected this typo accordingly
“In the Introduction there some unnecessary repetitions and incorrect simplifications (e.g. the ENS consists also from glial cells that outnumber neurons)”
Response: We corrected the paragraph as following. “These communications occur: directly, through the vagus nerve (VN), whose stimulation is mediated by microbial metabolites and neuro-hormones [1] released from the enteric nervous system (ENS), that controls bowel function even though it is completely separate from the CNS; the ENS is made up of interneurons, sensory neurons, motor neurons, glial cells and neurotransmitters [2]. Furthermore, neuroinflammation could be caused also by the production of immunogenic microbial endotoxins (such as LPS), that act both through a direct damage or through the activation of immune cells [1,5,6,].”
“In general the titles of figures should be modified (It should not be: “Figure one summarizes….)”
Response: We modified the titles of the figures as following:
Figure 1. The pathways involved in the microbiota brain-gut axis
Figure 2. Stroke risks factor influenced by gut dysbiosis
“Figure 1 needs to be corrected. The vagus nerve should not be presented as a key part of top-down signaling, since it comprises mainly afferent nerves (up to 90%). Please see Figure 1 in Lee et al., 2021 (ref. 22).”
Response: We modified the figure 1 accordingly, since we added a red arrow with the meaning of an afferent signaling, and modifying the colors (now red) of the bottom up signaling
“It is difficult to agree with the first statement of paragraph 2a (page 3) that “little is known about how the gut-brain axis changes with aging”. Actually, there are numerus data on age-related changes in gut microbiota characterized by its decreased stability and diversity”
Response: We apologize for our mistake and we erased the aforementioned statement in the revised and uploaded version of the manuscript.
“Page 3, paragraph 2a – Metabolic syndrome as a risk factor for ischemic stroke is associated with well characterized changes in gut microbiota composition that should be clearly presented.”
Response: Dear reviewer, as You suggested, we added the following paragraphs with regard to the complex interplay between gut microbiota and metabolic syndrome: “Several studies have shown that the composition of gut microbiota in healthy individuals was significantly different from that in obese individuals, which indicated that gut microbiota might play an important role in metabolic syndrome.For example, a study performed by Gordon and coworkers has highlighted significant differences in the microbiota composition of obese people by comparison with lean people [27-29]. Particularly, it has been demonstrated that a decrease in Bacteroides (Bacteroidetes phylum) and an increase of Bacillaceae, Clostridiaceae, and other representatives of the Firmicutes phylum in the gut of obese people [27,30,31] is directly connected to a prolonged exposure to a high-fat diet.Furthermore, it has been shown that intestinal microbiota affects body weight, lipid profile, and white adipose tissue both in mices and humans [32]. Among the mechanisms involved in weight regulation we report : the energy extraction from food, handling and storage, the beiging of the white adipose tissue, the role of indoles, and tryptophan-derived microbial metabolites in controlling adiposity via microRNAs in white adipose tissue [32]. Among the functional mechanisms involving the intestinal microbiota in lipid metabolism, we report: clearance and intestinal absorption of triglycerides, microbial-signals, as those involving bile acids and SCFA [32]. Furthermore, also the development of insulin-resistance is known to be regulated also by the interplay of different metabolites that influence insulin signaling and inflammatory processes [32]. Among these metabolites, produced by intestinal microbiota, we report imidazole propionate, tryptophan-derived metabolites, indoles, kynurenine, serotonin, branched-chain amino acids, and SCFA [32].”
“In paragraph 2b – “The effect of gut dysbiosis in influencing stroke outcomes” – the causal relationship between ischemic stroke and gut dysbiosis is not clearly presented. Potential mechanisms of stroke-induced dysbiosis should be highlighted, whereas the Authors are pointing to dysbiosis-induced intestinal paralysis and increased gut permeability. This is confusing in understanding of the cause and consequence effect.”
Response: We reported more extensively the major experiences regarding the influence of ischemic stroke on gut microbiota, as following: “The features of the gut microbiota in patients with ischemic stroke have been examined in clinical studies [61]In the study performed by Yamashiro and coworkers [58], Ischemic stroke was associated with increased bacterial counts of Atopobium cluster and Lactobacillus ruminis and decreased numbers of the Lactobacillus sakei subgroup, and gut dysbiosis was associated with changes in serum metabolic and inflammatory marker levels. Yin and coworkers found an increase in opportunistic pathogens (Enterobacter, Megasphaera, and Desulfovibrio) and decrease in commensal or beneficial genera including Bacteroides, Prevotella, and Faecalibacterium (butyrate acid producer) in patients with ischemic stroke [60], whereas Li and coworkers found an increase in SCFA-producing bacteria including Odoribacter, Akkermansia, Ruminococcaceae_UCG_005, and Victivallis in patients with ischemic stroke [52]. Finally, Tan and coworkers lack of SCFA-producing bacteria (Roseburia, Bacteroides, Lachnospiraceae, Faecalibacterium, Blautia, and Anaerostipes) and an increase in opportunistic pathogens (Enterobacteriaceae and Porphyromonadaceae) and Lactobacillaceae and Akkermansia in patients with ischemic stroke. In this study, reduced fecal acetate level was associated with an increased risk of 90-day poor functional outcomes [49]. Furthermore, other studies focused on dysbiosis-induced intestinal paralysis, an increased gut permeability and the loss of cholinergic innervation in ileum, and an increased sympathetic activity [1,5,40,62,63]
“Further, on page 6, the well know association between IBD and ischemic stroke is briefly highlighted, but nothing is mentioned about the role of gut dysbiosis in that association. Again, according to the main title, that should be the main focus of this narrative review.”
Response: We modified the text as following, trying to mention more exhaustively the role of gut dysbiosis: “In a recent study the authors found a reduction of SCFAs-producing bacteria in cases with ischemic stroke when compared to controls [80]. Nonetheless, a study showed that the butyrate-producing bacteria were remarkably less abundant in ischemic stroke and with increasing abundance of lactic acid bacteria [81]. In another study the authors investigated the effects on cellularity of the lymphoid tissue of the gut. In particular, they observed a reduction of T and B cells after ischemic stroke, and a stability in the amount of natural killer cells and macrophages [82]. A reduction in parasympathetic nerve activity following acute CNS injury is associated with overgrowth of bacteria in the bowel and increased bacterial translocation. Substances released by bacteria and metabolites and media released by the gastrointestinal immune system can affect gastrointestinal movement [83,84]. In particular, it has been described that, after CNS injury as ischemic stroke, mast cells present in the gut release histamine and tryptase activating submucosal neurons and peptidergic neurons to secrete vasoactive intestinal peptide (VIP) which alters the movements of the gastrointestinal tract [85]. Enteric nervous system with the release of nitric oxide can also lead to a delay of gastric emptying [86]. Recently a link between ischemic stroke and inflammatory bowel disease (IBD) has been proposed, although studies about the relationship between these two clinical entities are not currently available and the hypothesis of a link is mainly based on few observational studies. In particular, IBD patients present a higher risk for arterial and venous thromboem-bolism (TE) than patients without IBD and this complication is associated with a consid-erable morbidity and mortality rates (overall mortality 25% per episode). In particular, cerebrovascular TE represents the most frequent and severe CNS complication in IBD [87]. In a retrospective single center cohort study it has been demonstrated that the risk of TE is higher in patients with active IBD than IBD in remission [88]. In a population-based retrospective cohort study it has been shown that the frequency of IBD exacerbation and hospitalization are risk factors for ischemic stroke [89]. Available data regarding cerebrovascular risk modification induced by treatment for IBD are contradictory. A beneficial effect with increased carotid-femoral pulse wave velocity (PWV) has been shown with salicylates, but not with steroids or azathioprine. Moreover, TNF-a inhibitors appear to decrease the ischemic heart disease rate but increase the risk of cerebrovascular events [90]. In the association between IBD and ischemic stroke the role of gut dysbiosis seems to play a fundamental role. In particular in IBD patients has been described an expansion of potential pathogens such as Enterobacteriaceae includind E.Coli and changes in microbial composition with a reduction of Firmicutes species [91]. Moreover, the metabolic pathway of amino acid biosynthesis, carbohydrate metabolism, osidative stress and bile salt seem to be modified in the microbiota of patients affected by IBD, suggesting a functional impact of gut microbiota on pathogenesis of these diseases. [92] In IBD patients has been reported an aberrant immune response to microbial dysbiosis due to genetic alteration in innate immunity, intestinal barrier, microbial recognition and processing. It leads to a persistently stimulation of proinflammatory condition and macrophage/monocyte infiltration in the gut [93]. Some recent studies present an excellent example for the modulation by gut microbiota through gut‐brain axis via bottom-up in IBD related neurological complication such as memory impairment and anxiety-like behaviour in animal models [94-96]. Since IBD patients present higher risk of ischemic stroke, gut‐brain axis is very likely to represent a potential link between gut pathology and the increased risk of ischemic stroke. However, currently there are still no studies in the literature regarding the role of gut‐brain axis in the relationship between ischemic stroke and IBD. Moreover, there is no data about the potential role of IBD therapies on risk of ischemic stroke. Future studies are needed to establish causal association between active IBD and cerebrovascular events in order to also establish a possible therapeutic target.”
“Table 1 distinctly demonstrates lack of flow of the review. It consists of two separate parts: 1) the role of microbiota-gut-brain axis in stroke related gastrointestinal symptoms and 2) link between IBD and ischemic stroke. By the way, it is not clear why only 4 papers were selected for the first part of the table, since there are much more data on that topic – see ref. 24”
Response: We rebuilt the table 1 as You suggested
Finally, we corrected all the references and the manuscript was revised by a native English speaker from MDPI.
Many thanks again
Sincerely Yours
Emanuele Sinagra
Reviewer 2 Report
Re: Manuscript ID: life-1258473.
This is a short review dealing with the involvement of the microbiota-gut-brain axis in inflammatory bowel disease. The work is a general overview on this argument. Changes are suggested to improve the paper.
Points of criticism
The expression “gut-brain-microbiota axis” must be replaced with “microbiota-gut-brain axis”.
The authors report that after brain injury gastrointestinal epithelium is disrupted. Do the authors have evidence of similar effects in other organs (respiratory or genitor-urinary) which also have a microbiota?
I suggest the authors to include a short comment on the possible involvement of blood brain barrier and glymphatic system.
Figure 1 is not mentioned in the text.
Page 2, line 15 from top.
This sentence must be aligned on the left margin without black point.
Page 2, line 8 from bottom.
Delete “of”.
Page 3, line 10 from top.
Full stop after [22].
Page 3, line 17 from bottom.
Delete “A”.
Page 3, line 9 from bottom.
Firmicutes/Bacteroidetes in italic.
Page 4, line 16 from top.
Replace “occur” with “occurs”.
Page 4, line 11 from bottom.
Firmicutes/Bacteroidetes in italic.
Page 5, line 8 from bottom.
Full stop after the reference. Check in many other lines.
Page 5, line 7 from bottom.
Replace “lead” with “leads”.
Page 5, line 5 from bottom.
Replace “lead” with “leads”.
Page 5, line 2 from bottom.
Peptococcaceae and Prevotellaceae in italic.
Page 6, line 18 from top.
Replace “alter” with “alters”.
Page 6, line 20 from top.
Replace “led” with “lead”.
Page 6, line 29 from top.
Replace “represent” with “represents”.
Page 6, line 29 from top.
Replace “complications” with “complication”.
Page 6, lines 35-36 from top.
Avoid repetition of “current” and “currently”.
In table 1 Peptococcaceae and Prevotellaceae in italic.
Page 7, line 12 from bottom.
Replace “appear” with “appears”.
References are not typed according to journal guidelines.
Author Response
Response to reviewers’ comments
Dear Editors, dear Reviewers,
We wish to express our appreciation to the Editors and Reviewers for their insightful comments, which have helped us significantly to improve our manuscript. According to the suggestions, we have thoroughly revised our manuscript and its final version is enclosed. Point-by-point responses to the comments are listed below.
Reviewers’ comments #2
“The expression “gut-brain-microbiota axis” must be replaced with “microbiota-gut-brain axis”.”
Response: We used the expression: “microbiota-gut-brain axis” with dashes in the whole manuscript, as uploaded
“The authors report that after brain injury gastrointestinal epithelium is disrupted. Do the authors have evidence of similar effects in other organs (respiratory or genitor-urinary) which also have a microbiota?”
Response: We stated as following:
The mechanism underlying immune dysregulation and reduced antimicrobial defense after acute ischemic stroke is still unclear [22]. There is evidence of bacterial translocation via paracellular means post-stroke (ZO-1 tight junction breakdown) at the gut level, even if there are no data about an eventual epithelial disruption in other systems (as pulmonary or genito-urinary) which also have a microbiota.
“I suggest the authors to include a short comment on the possible involvement of blood brain barrier and glymphatic system.”
Response: as You suggested, we added the following paragraph: “Post-stroke loss of blood-brain barrier (BBB) integrity influences perivascular space integrity and glymphatic system efficiency [64-67]. Several studies, in animal models as well as in patients, have shown that impaired cerebrospinal fluid (CSF) and glymphatic clearance could contribute to accumulation and aggregation of waste products and other compounds in the CSF and brain parenchyma [64,68]. Therefore, neuroprotective measures are crucial in order to prevent neuronal death and to avoid an increase of the BBB breakdown area [64].”
“Figure 1 is not mentioned in the text”
Response: We corrected this error by adding the mention in the text.
Finally, we corrected all the references and the manuscript was revised by a native English speaker from MDPI, in order to raise all the issues highlighted by both the reviewers.
Many thanks again
Sincerely Yours
Emanuele Sinagra

Round 2
Reviewer 1 Report
I appreciate the Authors’ effort to address all the remarks and suggestions in the revised version of the paper.
However, the main concern remains the same: the title “The role of the microbiota-gut-brain axis in ischemic stroke: a narrative review with a focus on its relationship with inflammatory bowel disease” does not reflect the content of the paper. In the revised version (page 8) the Authors stated: “However, currently there are still no studies in the literature regarding the role of gut‐brain axis in the relationship between ischemic stroke and IBD.” In light of that statement the title requires modification since this paper is not on the microbiota-gut-brain axis in ischemic stroke WITH A FOCUS on inflammatory bowel disease.
Major comments:
- There is a confusing statement on page 7: “Recently, a link between ischemic stroke and inflammatory bowel disease (IBD) has been proposed, although studies about the relationship between these two clinical entities are not currently available and the hypothesis of a link is mainly based on few observational studies.” This is not true and not consistent with Tab. 1, in which studies on the link between ischemic stroke and IBD are cited (in fact there are more than two studies regarding that topic).
- Page 7, The statement “In the association between IBD and ischemic stroke the role of gut dysbiosis seems to play a fundamental role.” is speculative.
Additional minor remarks:
1. There still some language issues requiring polishing:
- the title of Figure 2 “Stroke risks factor influenced by gut dysbiosis” – it should be rather “Stroke risk factors influenced by gut dysbiosis”
- please do not overuse the wording “we report” – page 4
2. While mentioning Gordon et al. the following references are given 27-29 – please check ref. 27
Author Response
Dear REviewer
Dear Editors, dear Reviewers,
We wish to express our appreciation to the Editors and Reviewers for their insightful comments, which have helped us significantly to improve further our manuscript. According to the suggestions, we have thoroughly revised our manuscript and its final version is enclosed. Point-by-point responses to the comments are listed in attachment.

Reviewer 2 Report
The authors amended the reviewer's recommendations.
Author Response
Dear Editors, dear Reviewers,
We wish to express our appreciation to the Editors and Reviewers for their insightful comments, which have helped us significantly to improve further our manuscript. According to the suggestions, we have thoroughly revised our manuscript and its final version is enclosed. Point-by-point responses to the comments are listed as attachment.
